# Living with depression and diabetes: A qualitative study in Bangladesh and Pakistan

**Hannah Maria Jennings**[1,2]*, **Ashraful Anas**[3], **Sara Asmat**[4], **Anum Naz**[5], **Saima Afaq**[4], **Naveed Ahmed**[3], **Faiza Aslam**[5], **Gerardo Zavala Gomez**[1], **Najma Siddiqi**[1,2], **David Ekers**[1,6]

1 Department of Health Sciences, University of York, York, United Kingdom, 2 Hull York Medical School, Heslington, United Kingdom, 3 Diabetic Association of Bangladesh, Dhaka, Bangladesh, 4 Institute of Public Health and Social Sciences, Khyber Medical University, Peshawar, Pakistan, 5 Institute of Psychiatry (IoP), Benazir Bhutto Hospital, Rawalpindi Medical University, Rawalpindi, Pakistan, 6 Tees Esk and Wear Valleys NHS FT, Darlington, United Kingdom

* hannah.jennings@york.ac.uk

**Data Availability Statement:** The availability of the full data set is not publicly available in a repository due to ethical restrictions. Ethical approval for the

## Abstract

Diabetes and depression are both serious health conditions. While their relationship is bidirectional and each condition adversely affects outcomes for the other, they are treated separately. In low and middle income countries, such as Bangladesh and Pakistan, health systems are already stretched and the integration of diabetes and depression care is rarely a priority. Within this context through interviews with patients, healthcare workers and policy makers the study explored: lived experiences of people living with depression and diabetes, current practice in mental health and diabetes care and barriers and perspectives on integrating a brief psychological therapy into diabetes care. The findings of the study included: differing patient and practitioner understandings of distress/depression, high levels of stigma for mental health and a lack of awareness and training on treating depression. While it was apparent there is a need for more holistic care and the concept of a brief psychological intervention appeared acceptable to participants, many logistical barriers to integrating a mental health intervention into diabetes care were identified. The study highlights the importance of context and of recognising drivers and understandings of distress when planning for more integrated mental and physical health services, and specifically when adapting and implementing a new intervention into existing services.

## Introduction

### Diabetes and depression in Bangladesh and Pakistan

Both diabetes and depression pose challenges to individuals, society and health systems worldwide. Type 2 diabetes (T2DM) is one of the top five causes of years lived with disability [1]. T2DM is the third leading cause of mortality world-wide, almost 80% of cases which occur in low and middle income countries (LMIC) [2]. South Asia has faced a rapid rise in diabetes and it is now the single most prevalent non-communicable disease (NCD) [3]. Prevalence of diabetes is estimated at approximately 14% in Bangladesh [2] and almost 17% in Pakistan [4].

study was received from the Health Sciences Research Governance Committee at the University of York (chaired by Professor Stephan Holland stephen.holland@york.ac.uk). When applying for ethical approval we did not specify that the data would be made publicly available in a repository. As part of the written and verbal consent we assured participants that all data would be confidential and access to the recordings would be restricted to the research team. We did specify that "some of their words" may be used in reporting the findings of the study (which we have done within the manuscript as non-identifiable quotes), however to make all raw data publicly available will be a serious breach to the rights of ethical of participants given did not consent to this.

**Funding:** This work is part of the DiaDeM programme, funding was received from the National Institute for Health and Care Research (NIHR) Research and Innovation for Global Health Transformation (RIGHT) funding programme, grant number NIHR200806. NS is the PI of DiaDeM and HMJ, SA and DE are named co-investigators. All co-authors (HMJ, SS, AN, SA, NA, FA, GZG, NS and DE) received funding from this award. The funder had no role in the study design, data collection, analysis or writing of the paper.

**Competing interests:** The authors have declared that no competing interests exist.

Mental health conditions, including depression, cause suffering worldwide. South Asia accounts for approximately 20% of the world's disease burden due to mental illness [5]. Depression is the single largest cause of global disability according to the World Health Organisation [6], it is also associated with reduced life expectancy and adverse life events [5]. While estimates of depression prevalence stand at just over 4% in both Bangladesh and Pakistan [5], people with long-term health conditions are significantly more likely to suffer from depression [7, 8]. Diabetes and depression are bidirectional, with shared environmental and biological causes, and each condition adversely affects outcomes for the other [9, 10]. Biomedical health systems worldwide tend to treat the 'mind' and 'body' separately, hence the mental health of people with chronic physical conditions, such as diabetes, tends not to be addressed within their services and vice versa–however, there is an increasing recognition of the need for more integrated services [9, 11, 12]. In low resource settings where services are overstretched and under-staffed, the integration of mental and physical health services are not viewed as a priority [11].

## Behavioural activation in diabetes care

One way of integrating mental health care into physical health facilities is through training staff to deliver brief psychological therapies such as behavioural activation (BA). BA is a brief psychological intervention, that is based on the premise that changing behaviour and positive reinforcement leads to an improvement in mood, therefore treatment focuses on encouraging and sustaining behaviour change [13, 14]. BA has been demonstrated to be effective in moderate to severe depression [15], it has much potential as it is flexible, adaptable and can be implemented by non-specialists [16, 17]. It has also been suggested that it is less stigmatising than biological narratives of depression as its rationale is not based on inner causes–additionally it can be adapted to incorporate and consider one's culture and values [16].

Lehmann and Bördlein [14] conducted a review of evidence on the application and adaptation of BA for culturally diverse populations and minorities, finding that adapting BA to a population or groups' need would increase its accessibility and acceptability leading to increased adherence and better outcomes. Furthermore, BA is one of the recommended interventions to be used to tackle depression in low-resource settings [16]. An intervention that incorporated BA delivered by lay counsellors was found to be effective to treat depression in India, suggesting that BA can be adapted to the South Asia context [18, 19].

Given the potential of BA and the need for mental health care within diabetes care within low-resource settings such as Bangladesh and Pakistan, the DiaDeM (Developing and evaluating an adapted behavioural activation intervention for people with depression and diabetes in South Asia) programme is adapting and testing BA within diabetes care in Bangladesh and Pakistan. Prior to adapting and implementing any new intervention it is vital to understand the context in which it is being introduced. We therefore set out to explore people's lived experiences and current practices around depression in diabetes care in this context, as well as examining how psychological therapies (and BA specifically) are perceived. The study explored the views of people with diabetes and depression, health care workers and policy makers. The specific objectives of the study were:

1. To explore the lived experiences of people living with diabetes and depression in Bangladesh and Pakistan

2. To understand current practices regarding recognising and treating distress/depression in diabetes care

3. To explore views on whether a brief psychological therapy such as BA could be integrated into diabetes care in Bangladesh and Pakistan

## Material and methods

### Setting

The study is nested in the DiaDeM programme, an NIHR funded programme investigating the integration of BA into diabetes care in Bangladesh and Pakistan. This study provided important context for the programme and helped to inform the adaptation of BA. The four sites across Bangladesh and Pakistan where the research occurred are where the study is being implemented. In Bangladesh data collection took place in the capital (and largest city), Dhaka, and the country's fourth largest city in the southeast of the country, Sylhet. In Pakistan data collection took place in Rawalpindi, located in the Punjab and Pakistan's 4th largest city, and Peshawar, the 7th largest city and the capital of the Khyber Pakhtunkhwa (KP) province of Pakistan. Both countries are majority Muslim, although in Bangladesh there is a significant Hindu minority (around 10%). In Bangladesh 98% of the population speak Bangla and are Bengali, with Sylheti being a dialect of Bangla. Rawalpindi is majority Punjabi (approximately 84%), and Peshawar majority Pashtun (Around 90%) with a growing number of Afghan refugees. All four cities are hubs for specialist medical care in their region, with Sylhet and Peshawar particularly catering to rural as well as urban populations.

### Participants and recruitment

There were three categories of participants interviewed that needed to meet the following inclusion criteria: 1. patients aged over 18 diagnosed with depression and diabetes; 2. health care workers or managers that worked within health centres that worked with patients with diabetes and/or mental health difficulties; 3. People involved in contributing to local or national health policy. Participants were recruited purposively to ensure for a range of characteristics (mixture of age, gender and professions). Sites of recruitment for patients and healthcare workers were: BIRDEM (Bangladesh Institute for Research and Rehabilitation in Diabetes, Endocrine and Metabolic Disorders) hospital, Sylhet Diabetic Association hospital, Sughra Diabetic Centre, Benazir Bhutto hospital, the Institute of Psychiatry Rawalpindi, Peshawar 'sugar' hospital, Lady reading hospital and Hayatabad medical complex. Policy makers were recruited through recommendations and contacts. Researchers (AA, SA, AN) familiar with the sites, were responsible for approaching potential participants and introducing themselves and their role in the research. They explained the purpose of the research (with accompanying written information) and invited them to take part. If the potential interviewee was interested in taking part, they were contacted and if still interested an interview was arranged at a time convenient to them–either by phone or in person.

A total of 48 semi-structured interviews were conducted (n = 23 in Pakistan and n = 25 in Bangladesh) with people diagnosed with depression and diabetes (n = 18), health care workers/managers (n = 22) and policy makers (n = 8). People with diabetes included men and women with ages ranging from 32 to 62 years old. Health workers included a range of health professionals from diabetes care and mental health. Policy workers included recruits from mental health, NCD control, health systems and medical education. Table 1 describes the characteristics and number of interviews.

### Data collection

In-depth interviews with participants took place between March and June 2021. Interviews were conducted by researchers (AN, AA and AS) in-country in Bangla or Urdu. Interviews were guided by topic guides. Topic guides were based on the objectives of the research. We also drew on basic BA theory, i.e. the link between behaviour and mood leading to the focus

**Table 1. Characteristics of people interviewed.**

| | Bangladesh | Pakistan | Total |
|---|---|---|---|
| People with diabetes and depression | 5 women<br>3 men | 2 women<br>8 men | N = 18 |
| Health care workers and managers | 4 Manager/director<br>2 Mental health specialists/ psychiatrists<br>5 Clinicians/diabetes specialist/endocrinologists<br>1 Diabetes educator | 2 managers<br>2 Psychiatrist<br>6 Clinicians/diabetes specialist/endocrinologists | N = 22 |
| Policy makers | 2 Mental health<br>1 NCD control<br>1 Medical education<br>1 Health Systems/services | 3 Health systems/services | N = 8 |
| Total | N = 25 | N = 23 | **N = 48** |

on behavioural change in treatment [20]. In order to ensure this concept was understood an explanation of BA was developed and piloted in both Bangladesh and Pakistan, and shared with participants during the interview. The topic guide included questions about current practices around depression, and experiences of depression. There were questions covering the context, experiences and views on 'talking therapy'. Additionally questions were asked about the concept of BA specifically (with probes about the link between activities, behaviour and mood) as well as the practicalities of integrating it in practice from a care provider and receiver perspective and in terms of policy. Probes included asking for specific examples. The topic guides were piloted and translated into Bangla and Urdu.

Prior to the interviews informed consent was taken, and the interviewees were informed that they could withdraw from the study at any time. The interviews took place in a private location (in a room in the health centres or office space) and a carer/family member could also be present if the participant requested, participants were also given the option of a telephone interview if there were difficulties or concerns in being able to do a face to face interview (3 interviews were done by phone). Face to face interviews followed Covid-19 protocols (social distancing and wearing face masks). The interviews were audio-recorded and lasted approximately 30–60 minutes.

## Data analysis and management

The recordings of the interviews were transcribed and translated verbatim directly into English. Transcripts were anonymised, uploaded onto software NVIVO and shared securely on a private server within the research team. The data were analysed thematically [21, 22] by four researchers (AN, AA, SA, HMJ) following the six-step procedure: 1. Familiarising ourselves with the data, 2. Generating a coding framework, 3. Searching for themes, 4. Reviewing themes, 5. Defining and naming themes, 6. Generating a report. We had regular meetings where we discussed and finalised the coding framework. The data were then coded in NVIVO, before agreeing on themes. The subsequent analysis was done in two stages: 1. Themes were written up separately for the Bangladesh and Pakistan data by the researchers in each country; 2. The reports were then combined and interpreted together in order to refine the final results, this was led by HMJ in discussion with country teams.

All 4 researchers involved in the data collection and analysis work on the DIADEM programme (AA, AN, and SA as Research Fellows and HMJ is a lecturer) and are familiar with the context. AA is a man from Bangladesh and is a native Bangla speaker; AN and SA are women from Pakistan and are native Urdu speakers; all three are experienced health researchers and

have completed Masters degrees (AA, AN) or have a medical MBBS degree (SA). HMJ has worked in and with teams in South Asia for many years and has knowledge of Bangla. As a team we discussed the research and were able to check and reflect on any difficulties as well as our reflections and findings of the research. We followed the Consolidated Criteria for Reporting Qualitative Research (COREQ) for reporting the results of the study (S1 Checklist).

### Ethics

We considered the implications of our research for the participants, all research participants were provided verbal and written information about the research. Informed consent was taken. Ethics approval was granted by the University of York (HSRGC/2020/409/B), National Bioethics Committee Pakistan (NBC-578), Khyber Medical University (DIR/KMU/UEC/25), Rawalpindi Medical University (242/IREF/RMU/2020)IOP and the Diabetic Association of Bangladesh (BADAS-ERC/EC/20/00300) for the study as part of the wider DIADEM programme.

## Results

The participants accounts are organised according to three broad categories, each covering a number of themes. They include: 1. Experiences of depression (themes: understandings and drivers of depression; lack of acceptance and stigma about mental health; living and coping with distress); 2. Responses to depression (themes: 'usual care' for depression; care-seeking for depression); and 3. Views on BA (themes: changing activities as a way of alleviating mood; barriers and enablers to BA; NCD and mental health policy environment).

### Depression: Experiences, coping, care-seeking and treatment

**Theme 1: Understandings and drivers of depression.** While all the patient participants in the research did have a diagnosis of depression, unlike healthcare workers, they rarely (with some exceptions) used the term 'depression'. In Bangladesh, several participants reported a lack of understanding around the term 'depression' itself. Rather than referring to oneself as 'depressed' per se participants would discuss feelings of sadness. Additionally forgetfulness, anger, loneliness, irritation, despondence and a lack of interest and ability to do everyday activities were also described.

"*I do not understand anything, and I do not understand depression.*"

Woman in Bangladesh with lived experience of depression and diabetes, aged 51

"*Sometimes I want to be alone, sometimes I do not feel like talking to others, sometimes I get angry when people talk.*"

Man in Pakistan with lived experience of depression and diabetes, aged 36

A host of physical symptoms associated with feelings of distress were commonly described including: problems with sleep, restlessness, palpitations, dizziness, tremors, body pain, headaches and menstrual cycle disruptions. While most patient participants did describe at least some physical symptoms, there were considerable variations between individuals as to the type and extent of symptoms.

"*I feel like a burden has fallen on my brain. I have headaches, or pain in my body and that's how I came to know* [about depression]"

Man in Pakistan with lived experience of depression and diabetes, aged 48

"*I feel restless, my throat feels heavy, I get palpitations and I feel dizzy most of the time.*"

Woman in Bangladesh with lived experience of depression and diabetes, aged 32

The reasons for distress identified commonly included financial difficulties and chronic disease. Other reasons mentioned included family problems, past abuse, lack of freedom (for women) and unemployment. In one of the sites in Pakistan clear gender differences were identified–with men more likely to identify unemployment and financial issues as reasons for distress. When discussing family issues, it was recognised that being distressed could exacerbate family issues as well as cause them.

"*the most common reasons* [for distress] *are financial issues, unemployment, not being able to concentrate on my studies and issues at home*"

Man in Pakistan with lived experience of depression and diabetes, aged 36

Chronic disease was identified as a cause of distress, this included worries about health as well as associated costs. In one of the sites in Pakistan hypertension specifically was confused with depression and used interchangeably with it. When asked specifically about diabetes, while several participants did not explicitly identify a link between diabetes and mental health, many did. As well as the more general worries about health and associated costs, it was explained that as diabetes is a disease that has an impact on one's life daily it thereby causes distress. Participants too described that one's mood could impact on their diabetes and blood sugar. Some participants discussed depression and the impact on their physical health and chronic disease more generally.

"*Obviously, your diabetes is better when you are in a good mood. If I get depressed I will have high blood sugar; both of these diseases have a deep relationship with me.*"

Man in Pakistan with lived experience of depression and diabetes, aged 30

"*When I have low blood sugar or high diabetes, I feel tense and I feel sick. When I get tense I cannot sleep well and if I cannot sleep well I do not feel like doing anything all day. My chest starts throbbing and I do not feel like eating anything. For all these reasons, my diabetes is not normal.*"

Woman in Bangladesh with lived experience of depression and diabetes, aged 30

Healthcare workers in both Bangladesh and Pakistan did identify a relationship between diabetes and depression. In one of the health facilities in Pakistan healthcare workers explained patients with depression would often not report their symptoms, however having depression would negatively affect medication adherence and attendance to health appointments meaning their physical health (including diabetes) would deteriorate, which would in turn affect their mental health.

"*Therefore if they stay depressed their compliance in taking medication will be affected, they may not come for follow-up visits and ultimately their diabetes will get worse. And they will develop complications, and the misery of the patient will get worse*"

Health worker in Pakistan

Having a chronic disease, such as diabetes, and the complications and financial implications were said to contribute to people becoming depressed, which in turn could exacerbate their diabetes (as explained above).

"*A diagnosis of diabetes itself is depressing for patients. Patients start suffering from depression ever since their diagnosis*"

Health manager in Bangladesh

"*Diabetes is a chronic and lifelong disease. Diabetes complications and challenges faced during treatment leads to depression in diabetes patients*"

Healthcare worker in Pakistan

**Theme 2: Lack of acceptance and stigma about mental health.**    In both Bangladesh and Pakistan patients and health care workers reported stigma around discussing mental health, having a diagnosis and visiting mental health services or a psychiatrist. The reaction to a diagnosis of depression and support received from relatives were somewhat mixed. While family members do often try to look after and support their relatives, this was not always the case–additionally there was often silence about the mental health of the individual from the patient and a feeling that people would not understand and keep a distance from them.

"*My relationships have not been good with anyone since I have depression. People who were close to me have distanced me from their lives. Only my mother-in-law helps me*"

Woman in Bangladesh with lived experience of diabetes and depression, aged 30

"*No I have never talked to my friends about this thing* [depression]"

Man in Pakistan with lived experience of diabetes and depression, aged 59

Healthcare workers spoke about psychiatry being very stigmatised meaning people will avoid going to mental health services and will avoid any referrals to a psychiatrist if suggested by a physician. Having a diagnosis could also be difficult to accept.

"*Patients have a poor acceptance of depression. Patients accepts all their illnesses but not everyone accepts depression*"

Health care worker in Pakistan

"*Why should I consult a psychiatrist! Do you think I am crazy*?!"

Woman in Bangladesh with lived experience of diabetes and depression, aged 30

"*Mental health care in Bangladesh is still very much stigmatised*"

Mental health specialist in Bangladesh

**Theme 3: Living and coping with depression.**    As well as impacting one's health and diabetes specifically, peoples' mood (or depression) impacted their everyday lives, and particularly their relationships and work. Several participants reported being withdrawn and therefore not

engaging with people or everyday activities, with some participants (in Pakistan) described not being able to regulate their feelings of anger or hyperactivity at times.

"*It* [depression] *affected my activities, I don't feel like doing things like before*"

Man in Bangladesh with lived experience of depression and diabetes, aged 62

"*Depression negatively affects my everyday activities. When I feel depressed I do not interact with family and friends. I also do not attend any events, even at work I just want to go home*"

Man in Pakistan with lived experience of depression and diabetes, aged 30

Several participants reported that they now no longer worked, and their relationships with family members particularly were affected as they engaged less with them and/or felt their mood increased tensions with family; impacting home and work life. The two factors were often related, as a decrease in the family income adds strains to the whole family. Additionally some participants related their mood (or 'depression') to irregular work and financial problems (as opposed to vice versus).

"*I cannot give time to relatives like before. I find it hard to communicate with them, there is a communication gap with them; these is always an emptiness I feel*"

Women in Bangladesh with lived experience of depression and diabetes, aged 50

"*I do not have a good relationship with them* [family and friends] *just because of my depression. When I am depressed I will argue and it ruins the whole environment. People will stay away from me because of my behaviour*"

Man in Pakistan with lived experience of depression and diabetes, aged 35

Despite tensions and negative effects on their relationships, many of the research participants identified their family members as their main source of support. However, this does vary across families and members of the family who will support them. Several participants did not feel their family supported them adequately, while others would rely on one or two members to give them physical and psychological support (such as talking to them, taking them to the doctor and making sure they are looked after). In both Pakistan and Bangladesh participants specifically mentioned daughters as sources of support.

"*My daughters help me. . .. I occasionally ask my niece to come and talk to me as well. They support me in doing household chores like cooking, cleaning and washing clothes.*"

Woman in Bangladesh, aged 32

"*They* [family members] *usually try to make me relaxed and counsel me. They help me to maintain a good diet and also take care of my medications*"

Man in Pakistan, aged 35

While all patient participants had had some kind of medical assistance with depression at some point (see theme 5) they also spoke about other ways of coping. As reported above some found talking with family members helpful. A few participants in both Pakistan and Bangladesh reported finding comfort in religion and reciting prayers and/or reading the Quran

helpful. In Pakistan there were a couple of participants who reported consulting *pirs* (spiritual healers) and taking their medicine.

*Researcher*: *So what do you do to deal with these problems*?
*Interviewee*: *. . .I keep sitting all day or offer prayers. I also take medicine*
Woman in Bangladesh with lived experience of depression and diabetes, aged 51

"*For my depression I take support from a* pir *as I have a strong belief in spiritual and traditional medicine. So I take their help by getting a* ta'wiz (amulet) *and* wazifa *to help me reduce my mental stress. I also read Quranic verses which I recite, this helps with relieving my stress*"

Man in Pakistan with lived experience of depression and diabetes, aged 36

Other less mentioned ways of getting temporary relief from their low mood reported taking a walk, watching TV, talking with neighbours, doing activities such as sewing and eating and drinking.

*Researcher*: *What helps you feel better at that time* [when down/depressed]?
*Interviewee*: *I take medicine and I walk outside*
*Researcher*: *Anything else*?
*Interviewee*: *No nothing. . .I do have a sewing machine, actually I sew to help me forget my problems*
Woman in Bangladesh with lived experience of depression and diabetes, aged 30
*Researcher*: *How do you deal with your depression*?
*Interviewee*: *I smoke, I do exercise, I usually discuss it with my close friends whom I can trust. . .when I discuss my problems I can relax.*
Man in Pakistan with lived experience of depression and diabetes, aged 33

### Responses to depression: Healthcare for depression

**Theme 4: "Usual" care for depression: Healthcare worker perspectives.** When diagnosing and treating depression the healthcare workers and managers in Bangladesh and Pakistan (with the exception of one site) reported a lack of guidelines and protocols for diagnosis and managing depression. While one manager in Bangladesh reported that they thought there were some guidelines as to whether a patient could be treated for depression at their facility, this was not verified and all other healthcare workers and managers in Bangladesh reported none. However, in one of the health facilities in Pakistan it was reported by a couple of health care workers that they will draw on international scales (Hamilton, Beck, PHQ2 and PHQ9 were mentioned) and try to follow international guidelines in the management of depression; any adaptation work on the scales and the extent to implementation however was somewhat unclear. Most of the non-mental health workers across the sites reported they did not have adequate training in mental health–the doctors mostly relied on what training they had received as an undergraduate.

*We psychologists and psychiatrists trained to follow guidelines. Hospitals do not have their own guidelines. We follow international guidelines such as NICE guidelines or Mosley guidelines–but due to our cultural norms it is not always possible. We do not have a checklist or a protocol in front of us–but we tell our consultants and trainees to try to follow these international guidelines.*

Health worker in Pakistan

*Researcher*: *Do you follow any checklist or protocol for the patients* [regarding depression]?

*Participant*: *No, there is no such thing*

Health worker in Pakistan

"*To the best of my knowledge, there is no such thing* [protocol or checklists]. *General physicians acquire this knowledge from their MBBS degrees and from further studies and various seminars. Very few physicians do any other training on psychiatry. Psychiatrists are the only ones who train on psychiatry later and are able to follow these protocols*"

Manager in Bangladesh

At all the sites health workers reported that doctors may diagnose depression and treat the patients with anti-depressants and may offer 'counselling' (talking with the patient). If it was felt that the patient needed further input or their depression was 'severe' they would be referred to a psychiatrist, psychologists or tertiary services. As highlighted above (theme 2), patients however may be reluctant to attend these services. There were some differences across healthcare facilities. In one of the sites in Bangladesh referral was uncommon due to a lack of mental health services in the area. More formal counselling and CBT was mentioned at one of the hospitals in Pakistan when referred to a psychologists.

"*First I will try to solve their problems, but when I feel that the patient needs proper treatment I refer them to a psychiatrist who can give better treatment and will counsel them in a proper way*"

Health care worker in Pakistan

"*We give drugs to a group of these patients and when the case is severe, we refer them to a psychiatrist in this hospital*"

Health care worker in Bangladesh

Consistently across the healthcare facilities, health workers and managers reported a need for improvement in practice particularly for mental health care. It was reported that there is a lack of training, guidelines and diagnostic tools. Within diabetes centres participants reported a dependence on medication, poor awareness among patients of mental health problems and a lack of time with the patients due to staff shortages and busy workloads. In Bangladesh the participants spoke about weak referral systems and a lack of mental health facilities, particularly outside of the capital Dhaka. Diabetes centres which patients usually attend for their check-ups and medicines are often busy with clinicians having limited time with their patients–reportedly just a few minutes per patient. Additionally, patients in the research highlighted time and costs associated with travel to the centres, as well as the high costs of medications (see theme 5).

"*Because there are a smaller number of doctors there is a huge load of patients. That is why it is difficult to treat patients with mental stress*"

Health worker in Pakistan

**Theme 5: Care-seeking for depression: Patient perspectives.** The process of receiving a diagnosis of depression was somewhat unclear from the interviews. In Bangladesh some participants reported their depressive symptoms to a doctor and subsequently received a

diagnosis, one patient reported being referred on to a psychiatrist who diagnosed their depression. In one site in Pakistan around half of the patients interviewed said they had self-referred themselves to the psychiatry department–others had been referred by a professional. In the other site in Pakistan most patients indicated their depression had been diagnosed by diabetes specialists. In Bangladesh, and to some extent in Pakistan, getting a diagnosis appeared to often be wrapped up in more 'physical' symptoms and seeking care for a variety of health conditions—during this process patients could at some point be diagnosed with depression from the medical professionals or be referred to psychiatry.

*Interviewee*: *There are many symptoms. I cannot sleep and my chest throbs. I do not like working, I feel restless all the time and nothing feels good.*

*Researcher: Have you visited any physician or health worker for these problems*?

*Interviewee*: *I went to a physician and he said I have this problem* [depression]. *Since then, I have been taking drugs for depression.*

Woman in Bangladesh with lived experience of depression and diabetes, aged 30

"*I would get angry easily for no reason, and I would get pressure on my eyes and shoulders; so I consulted a doctor and he told me I have depression and prescribed me anti-depressants*"

Man in Pakistan with lived experience of depression and diabetes, aged 35

In terms of medical treatment, the majority of patient participants (consistent with the health worker interviews) did report being on medication for depression. Across all sites other treatments (such as counselling or talking therapies) were not offered (with the exception of one health centre). Types of anti-depressant and patient concordance were not discussed, discussion was limited to whether medication should be taken. Most participants reported medication not being their only means of coping with depression (see theme 3), however it was recognised it was one way of coping with depression.

"*But medicines which I am taking for depression are making me quite calm*, *it really makes a difference*"

Man in Pakistan with lived experience of depression and diabetes, aged 59

"*No they do not do anything in particular* [about depression and diabetes]–*they just give medicines*"

Man in Bangladesh with lived experience of depression and diabetes, aged 62

There was a general consensus in both Bangladesh and Pakistan that very little attention was given to mental health by staff at diabetic centres. Even with a diagnosis of depression, most participants said they were rarely asked about mental health. In Bangladesh, two patient participants in one site did report being asked about mental health but there was little follow-up or additional treatment or therapy.

*Researcher*: *Now in addition to your depression or long-term discomfort, what kind of drugs, advice or what kind of help have you ever received from the diabetes centre*?

*Interviewee*: *No, I did not receive anything.*

Man in Bangladesh with lived experience of depression and diabetes, aged 51

*Researcher: Do you get any ongoing help or support from diabetes services with your depression*?

*Interviewee*: *Not often; usually I am so depressed and I start crying and then the doctor counsels me, but I think if this counselling was done on a regular basis it would be very effective and helpful.*

Woman in Pakistan with lived experience of depression and diabetes, aged 42

### Views on behavioural activation

**Theme 6: Changing activities as a way of alleviating mood.**    As part of the interviews an explanation of BA was given, and the participants were asked about the link between mood and behaviour. In theory participants generally agreed that there was a link between mood and behaviour–with a low mood meaning one is less likely to engage in everyday activities. Many of the patient participants reported activities they no longer engaged in or enjoyed. Health workers also agreed that a change or increase in activities would improve mood.

*Yes, I feel bad and restless, I cannot do household work. You know, I cannot even remember about cooking my lunch even it was 4 o'clock*

Woman in Bangladesh with lived experience of depression and diabetes, aged 32

*Yes of course my behaviour has changed and this has made my life miserable. I do not take interest in routine activities, I usually do not go to gatherings*

Man in Pakistan with lived experience of depression and diabetes, aged 50

*They will start to find solutions* [if they do BA] *and they will feel better with these activities. And those who have an element of depression and do not remember to take medicine will also improve and the patient will feel better.*

HW-02-PKR, health worker in Pakistan

Participants generally liked the idea of BA. Several said that they would like it to help both their mood and diabetes. In one site in Pakistan participants said they would like to have an opportunity to discuss their problems, while at the other site participants said they would appreciate motivation and encouragement from health workers.

*My bad mood will disappear and my diabetes will be under control. If Allah* [God] *wishes, then I can live long*

Man in Bangladesh with lived experience of depression and diabetes, aged 67

*A person who is close to you can motivate you. . .These types of things encourage someone to participate* [in activities].

Man in Pakistan with lived experience of depression and diabetes, aged 33

**Theme 7: Barriers and enablers to BA.**    The positive reaction to BA from patients and health workers is the main enabler to the integration of BA into diabetes care identified. People with diabetes and depression liked the idea of being seen individually by a health care worker and for a period of time. They saw BA as a way of improving their mood and health (and for some diabetes). Health workers were optimistic that BA could help patients to improve their diabetes as well as mental health and reduce dependence on medication and improve their

quality of life. In Bangladesh, health care workers thought that BA could bring positive change to patients through relieving depression, increasing self-confidence, and maintaining regularity in their daily life. Additionally as the therapy was perceived to be cost-effective and easy to access, it was thought to be more acceptable to the patients.

> *It [BA] will have effect on their behaviour as well as play an active role against diabetes*
>
> Health care worker in Pakistan

> *If it can be given effectively then it can definitely influence the patient and the way it will be given will increase the patient's self-confidence. So it will be effective.*
>
> Healthcare worker in Bangladesh

Despite the positive reaction to BA itself many barriers to the integration of BA in diabetes care were identified. From patient perspectives, in Bangladesh several mentioned the costs and time of travelling for a BA session–a couple of women mentioned finding it difficult to travel due to physical problems. Some mentioned they may not have the time to attend BA sessions. Participants in both countries also raised concerns about the crowded conditions, lack of private space for the sessions or staff capacity to conduct the sessions.

> "*I have problems regarding travelling and there is a cost issue as well*"
>
> Woman in Bangladesh with lived experience of depression and diabetes, aged 50

> "*There is one clinic for four to five days and six hundred patients visit the clinic daily. So what would doctors do in that situation? There is only one doctor sitting there. You can ask anyone.*"
>
> Man in Pakistan with lived experience of depression and diabetes, aged 59

Not having space and the resources to deliver BA in busy diabetic centres was also raised as an issue by healthcare workers, although in one of the sites in Pakistan some staff thought there was private space to conduct sessions. In Pakistan, healthcare workers highlighted the importance of administrative and managerial support for BA be implemented–they reported this is vital to ensuring there is both the budget to implement BA and that health workers are properly trained to deliver BA. In one of the sites the issue of changing practitioners' mindset to consider mental health and therapies as opposed to only considering medication was highlighted as a barrier.

> *Researcher: What challenges and opportunities may there be to deliver BA?*
> *Participant: Affordability, acceptability, time and transport issues should be taken into consideration.*
>
> Health care worker in Pakistan

> *First, there is no space for administration of BA. Second there is the patients' load over here. A third of people will not cooperate in it. Also it will only be possible after permission from the head of department and other administrative approval. It is not that easy to install it over here.*
>
> Health care worker in Pakistan

**Theme 8: NCD and mental health policy environment.** In Bangladesh policy makers identified a number of NCD policies and programmes, as well as a strategic plan for mental

health (not yet finalised)–hence they are on the country-wide agenda. The integration of mental health into NCD care, while often acknowledged to be important, is somewhat limited and specifically depression and diabetes care there was no evidence of integration. Furthermore policy makers interviewed highlighted problems of a lack of funds, staff, training, poor implementation of programmes, a lack of awareness among patients and an insufficient inclusion of common mental disorders in the curriculum of undergraduate medical training. However, they also identified positives of some training programmes and awareness raising programmes and the government's initiatives in establishing community clinics in remote parts of the country. The integration of depression care (including psychological therapies such as BA) was not contrary to any policy agenda–however implementation of any new intervention can take time and would need to be shown to be cost effective.

> "*The matter is at our policy level but implementation does not necessarily take place accordingly. As a result, when you give importance to an issue and include it at policy level the outcome at implementation level is not always favourable*"

Policy maker in Bangladesh

> "*Mental health has been linked to non-communicable diseases. Diabetes is a part of non-communicable disease and depression is a part of mental health. There is no separate policy for the two diseases. There is no such policy of depression and diabetes in our country. I do not know if it is present in other countries*"

Policy maker in Bangladesh

Pakistan has a provincial health system, hence the policy environment is different in the two sites of research in Pakistan. It was reported in Islamabad federal area the importance of integrating mental health into NCD care was agreed at a policy level, however policy makers said it could take time to happen and was currently limited. It was highlighted that the package of essential services now includes mental health which is a positive. While NCDs are an increased priority, issues of resources and funding were raised, as well as ongoing and competing health problems in Pakistan such as communicable diseases. Additionally, at the time of the research all resources and priorities were being directed to manage the Covid-19 pandemic, neglecting other areas in health. In Peshawar the policy makers did highlight the importance of integrating mental health with NCD care and were supportive of it in principle–however, there are currently no specific policies for mental health in the region. While this was an area in progress there remain budget and administrative constraints, as well as needing the support of health workers. The policy makers however could see the benefits and value in theory of integrative care.

> "*I can't tell you that when it [integration of mental health and NCD care] will happen and who will do it and how it will happen and where it will happen, but like a 100 other issues in Pakistan, they should also happen. We need to make it possible because the need is there and I think the good starting point can be that those who treat diabetes must be trained to see signs and symptoms of depression among patients who are presenting with diabetes*"

Policy maker in Pakistan

> "*Yes it [mental health and NCD care] should be integrated, as I mentioned mental health care and NCDs these two are the top priorities for the health department to incorporate along*

*with universal health coverage to provide basic minimum health standards to the community*"

Policy maker in Pakistan

## Discussion

The findings from this study show that depression is viewed differently by people diagnosed with depression and healthcare workers, while the latter unsurprisingly express a more medicalised understanding of depression the former discuss "distress" in terms of the everyday effects and behaviour. They also highlight different ways of dealing with depression which includes medication, religion, engaging in activities and social support. Both patients and healthcare workers agreed that mental health care is lacking and there needs to be more training for practitioners. Furthermore, both groups did generally recognise a link between depression and diabetes. In both Bangladesh and Pakistan personal experiences of any kind of therapy were very limited. However, in principle people with diabetes and depression were open to it and both patients and healthcare workers acknowledged the importance of addressing mental as well as physical health. Patients reported they would value more time with a healthcare workers (for both physical and mental health). During the interviews participants were given an explanation of BA, all groups reported that BA could be a logical and helpful way of addressing depression. People with diabetes and depression gave specific examples of changes in behaviour (or a lack of motivation to participate in activities) related to mood, which largely concurs with BA theory suggesting it is acceptable. From a policy perspective the integration of mental health and diabetes care (or NCD care) more generally is consistent with national plans. However, all three groups of participants expressed concern over the logistics of integration–namely a lack of staff, time and space; as well as having managerial support for any changes. From a patient perspective there were additional concerns about having time and money to attend additional appointments. The study gives us important insights into the experiences of people with depression and diabetes as well as some of the key considerations that need to be made when adapting a BA intervention.

The explanations of distress and understanding depression specifically, by those with a diagnosis of depression fall largely outside of a medical model of distress, and are instead tied up in the everyday experiences and are explained in terms of the wider structural and social causes of mental health problems. The importance of integrating local understandings and idioms of distress has been well researched [23, 24]. In the South Asian context, it has been suggested that terms such as "tension" commonly used can also be less stigmatising than labelling someone with an illness such as depression [25, 26].

While not seeking mental health care is commonly attributed to a lack of understanding or 'mental health literacy' the importance of the structural and social factors in explaining depression is increasingly acknowledged as important [27, 28]. Furthermore, a criticism of individual medical and psychological treatment is the decontextualising of distress from the wider environment [26, 28, 29]. While psychological therapies (including BA) and the clinical setting are indeed somewhat limited in the extent they can address these concerns, an acknowledgement of the lived experiences of depression and the wider socioeconomic context in understandings of depression is perhaps a start. Furthermore, embedding an intervention within a health system as well as making linkages with outside actors (such as communities or non-governmental activities) will mean it is more holistic [26].

As well as highlighting the structural and social reasons for distress, the participants in this study did make a link between their distress and diabetes. Other studies in South Asia have

found that most people who screen positive for depression do seek healthcare, often for physical rather than psychological reasons [30, 31], our results largely concur with this. While it is often assumed that patients do not understand the link between mental and physical health, our findings do not support this suggesting instead that people will understand the need to address mental health in a physical health setting. Furthermore, by integrating depression care into physical health it is plausible that it would be less stigmatising than seeking care only for depression. Indeed the stigma around mental health, and specifically seeking care from a mental health professional, was highlighted in the research. It has been suggested that BA could be less stigmatising than medication and other treatments [32], by integrating this into wider diabetes care the argument could be strengthened.

While the findings of the research highlight the need for integrated care, and that BA may be acceptable, the huge logistical constraints and potential for added work for an already over-stretched service were highlighted as a major barrier to the introduction of a new intervention. Indeed, in an earlier study some of the authors were involved in (HJ, FA) looking at the potential for BA in NCD care we considered it somewhat premature to integrate BA in this setting given the current restraints on health systems in Bangladesh and Pakistan [33]. We maintain this will be a major challenge. If any form of mental health care is to be integrated into physical health care it is important to work closely with specialist centres (such as diabetes centres) and consider practical steps such as aligning diabetes therapy consultations and carefully considering training and incentives. Longer-term should any form of mental health healthcare (including BA) prove effective in this setting it could help with improving outcomes for both diabetes and depression hence helping with the huge need in this setting.

It is important to consider the strengths and limitations of the current study. A strength of the study was that we were able to recruit a range of participants (patients, healthcare workers and policy makers) across two different countries and four sites. Some potential limitations of the study include the balance of participants. Among patient participants we recruited more men (N = 11) than women (N = 7), this were due to there being fewer women available to be interviewed in Pakistan. However, we did ensure there was representation from both genders across sites. Among the health professionals interviewed there were more participants who specialised in diabetes and/or worked in a diabetes centre (N = 16) than mental health specialists (N = 4). However, this is reflective of the scarcity of mental health specialists in both countries, additionally the participants reported that they were unlikely to visit or be diagnosed with depression by a mental health specialists. The study did not explore the use of digital technology in delivering BA, this was a missed opportunity given the challenges with transportation and access to care. However, we are exploring this aspect of BA as well as looking more in depth at the feasibility and acceptability of BA in a feasibility trial [34]. Having a relatively large number of in-depth interviews (N = 48) asking about experiences of physical and mental health and care from a range of perspectives provided rich data, was the main strength of the study.

## Conclusion

People in low resource settings living with diabetes and depression experience significant challenges in addressing their physical and mental health needs. There is a clear need for more holistic and integrated care addressing mental and physical health needs. While there are logistical challenges for integrating mental health care into physical health settings, working closely with institutions and having management support is essential as is ensuring any adapted interventions recognises the drivers and understandings of distress.

## Supporting information

**S1 Checklist. COREQ (COnsolidated criteria for REporting Qualitative research) checklist completed for the reported study in the manuscript.**
(PDF)

## Acknowledgments

We are grateful to the patients, healthcare workers and policy makers who gave us their time to participate in the interviews. We would also like to acknowledge the whole DiaDeM research team.

## Author Contributions

**Conceptualization:** Hannah Maria Jennings, Sara Asmat, Saima Afaq, Faiza Aslam, Najma Siddiqi, David Ekers.

**Data curation:** Ashraful Anas, Sara Asmat, Anum Naz.

**Formal analysis:** Hannah Maria Jennings, Ashraful Anas, Sara Asmat, Anum Naz.

**Funding acquisition:** Hannah Maria Jennings, Saima Afaq, Najma Siddiqi, David Ekers.

**Investigation:** Hannah Maria Jennings, Ashraful Anas, Sara Asmat, Anum Naz, Saima Afaq, Naveed Ahmed, Faiza Aslam.

**Methodology:** Hannah Maria Jennings, Ashraful Anas, Sara Asmat, Anum Naz, Saima Afaq, Naveed Ahmed, Faiza Aslam.

**Supervision:** David Ekers.

**Writing – original draft:** Hannah Maria Jennings.

**Writing – review & editing:** Hannah Maria Jennings, Ashraful Anas, Sara Asmat, Anum Naz, Saima Afaq, Naveed Ahmed, Faiza Aslam, Gerardo Zavala Gomez, Najma Siddiqi, David Ekers.

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
