## [Decision Letter · Decision Letter 0]

29 Mar 2023

PGPH-D-22-01710

Depression and diabetes in Bangladesh and Pakistan: a qualitative study exploring the feasibility and acceptability of integrating Behavioural Activation for depression into diabetes care

Dear Dr. Jennings,

Thank you for submitting your manuscript to PLOS Global Public Health. After careful consideration, we feel that it has merit but does not fully meet PLOS Global Public Health’s publication criteria as it currently stands. Therefore, we invite you to submit a revised version of the manuscript that addresses the points raised during the review process.

We look forward to receiving your revised manuscript.

Kind regards,

Abhijit Nadkarni

Academic Editor

Journal Requirements:

2. Your manuscript is missing the following sections: Introduction. Please ensure these are present, and in the correct order, and that any references to subheadings in your main text are correct. An outline of the required sections can be consulted in our submission guidelines here:

https://journals.plos.org/globalpublichealth/s/submission-guidelines#loc-parts-of-a-submission

4. In the online submission form, you indicated that "Data may be available upon reasonable request to the Principle Investigator of DiaDeM (NS) and lead for this study (HMJ)". All PLOS journals now require all data underlying the findings described in their manuscript to be freely available to other researchers, either 1. In a public repository, 2. Within the manuscript itself, or 3. Uploaded as supplementary information.

Additional Editor Comments (if provided):

The manuscript will require a major re-working to address the fundamental issues raised by Reviewer 1

Reviewers' comments:

Reviewer's Responses to Questions

**Comments to the Author**

1. Does this manuscript meet PLOS Global Public Health’s publication criteria? Is the manuscript technically sound, and do the data support the conclusions? The manuscript must describe methodologically and ethically rigorous research with conclusions that are appropriately drawn based on the data presented.

Reviewer #1: No

Reviewer #2: Yes

2. Has the statistical analysis been performed appropriately and rigorously?

Reviewer #1: N/A

Reviewer #2: N/A

3. Have the authors made all data underlying the findings in their manuscript fully available (please refer to the Data Availability Statement at the start of the manuscript PDF file)?

Reviewer #1: No

Reviewer #2: No

4. Is the manuscript presented in an intelligible fashion and written in standard English?

Reviewer #1: Yes

Reviewer #2: Yes

5. Review Comments to the Author

Reviewer #1: The authors have conducted a qualitative study in Pakistan and Bangladesh to explore experiences of depression and diabetes and to assess the feasibility and acceptability of a behavioural activation for depression in the diabetes treatment. The title as well as the objectives suggest that the main focus of the study is the feasibility and acceptability of BA intervention in this context. However, a large portion of the section is dedicated to describing the experience of depression in diabetes. The way the verbatim quotes have been presented and results handled in this section is also vague and non specific to the lived experiences. The feasibility and acceptability of the behavioural activation have not been explored in detail and have been presented superficially. The feasibility is presented in terms of space availability and resource availability issues. Acceptability has been explored very superficially. Given these limitations, I believe that the study has not achieved the objectives that it proposed.

Reviewer #2: This paper reads extremely well. All sections are well written and the data collection and analysis were well executed. It would be helpful for the authors to consider dropping the aim and objectives - and rather merge it with the introduction. A table describing who was recruited and their demographics would have been helpful.

Line 408 is unclear - it would be helpful to recheck. It is not clear whether the psychologist and psychiatrist are trained or the providers are trained by the duo. Line 544 - it would be helpful to provide in addition an alternative name of Allah.

It would have been helpful to explore the use of digital technology in delivering BA given the challenges with transportation etc.

6. PLOS authors have the option to publish the peer review history of their article (what does this mean?). If published, this will include your full peer review and any attached files.

**Do you want your identity to be public for this peer review?** For information about this choice, including consent withdrawal, please see our Privacy Policy.

Reviewer #1: No

Reviewer #2: No

---

## [Decision Letter · Decision Letter 1]

4 Jul 2023

PGPH-D-22-01710R1

Depression and diabetes in Bangladesh and Pakistan: a qualitative study exploring the feasibility and acceptability of integrating Behavioural Activation for depression into diabetes care

Dear Dr. Jennings,

Thank you for submitting your manuscript to PLOS Global Public Health. After careful consideration, we feel that it has merit but does not fully meet PLOS Global Public Health’s publication criteria as it currently stands. Therefore, we invite you to submit a revised version of the manuscript that addresses the points raised during the review process.

EDITOR: I agree with the reviewer that a large part of the content does not have to do with the acceptability and feasibility of BA. I would suggest you revise the manuscript to add relevant content about acceptability/feasibility or reframe the focus of the paper so that it encompasses the actual content of the manuscript.

We look forward to receiving your revised manuscript.

Kind regards,

Abhijit Nadkarni

Academic Editor

Journal Requirements:

Additional Editor Comments (if provided):

Reviewers' comments:

Reviewer's Responses to Questions

**Comments to the Author**

1. If the authors have adequately addressed your comments raised in a previous round of review and you feel that this manuscript is now acceptable for publication, you may indicate that here to bypass the “Comments to the Author” section, enter your conflict of interest statement in the “Confidential to Editor” section, and submit your "Accept" recommendation.

Reviewer #1: (No Response)

Reviewer #2: All comments have been addressed

2. Does this manuscript meet PLOS Global Public Health’s publication criteria? Is the manuscript technically sound, and do the data support the conclusions? The manuscript must describe methodologically and ethically rigorous research with conclusions that are appropriately drawn based on the data presented.

Reviewer #1: Partly

Reviewer #2: Yes

3. Has the statistical analysis been performed appropriately and rigorously?

Reviewer #1: No

Reviewer #2: N/A

4. Have the authors made all data underlying the findings in their manuscript fully available (please refer to the Data Availability Statement at the start of the manuscript PDF file)?

Reviewer #1: Yes

Reviewer #2: Yes

5. Is the manuscript presented in an intelligible fashion and written in standard English?

Reviewer #1: Yes

Reviewer #2: Yes

6. Review Comments to the Author

Reviewer #1: Thanks to the authors for making some changes. Unfortunately, large parts of the qualitative analysis still cover experiences of living with diabetes and depression, coping styles and treatment seeking behaviours. The feasibility and acceptability aspects of BA are not the main ideas in the paper even now. In the responses, the authors have replied that they disagree with the fact that feasibility and acceptability have not been explored well. However, I still do not find these aspects covered adequately in the paper. What are the barriers and facilitators for acceptable of BA methods. What are the experiences with implementing BA methods? These are not at all covered adequately even now. I dont think the paper convinces the reader that BA is feasible and acceptable to the community in Pakistan and Bangladesh.

Reviewer #2: No additional comments.

7. PLOS authors have the option to publish the peer review history of their article (what does this mean?). If published, this will include your full peer review and any attached files.

**Do you want your identity to be public for this peer review?** For information about this choice, including consent withdrawal, please see our Privacy Policy.

Reviewer #1: No

Reviewer #2: No

---

## [Editor Report · Decision Letter 2]

3 Jan 2024

Living with Depression and Diabetes: a qualitative study in Bangladesh and Pakistan

PGPH-D-22-01710R2

Dear Dr Jennings,

Thank you for revising the manuscript in response to our feedback. The framing of the paper is now consistent with the aims and methods.

We are pleased to inform you that your manuscript 'Living with Depression and Diabetes: a qualitative study in Bangladesh and Pakistan' has been provisionally accepted for publication in PLOS Global Public Health.

Best regards,

Abhijit Nadkarni

Academic Editor